# Citric Acid Assisted Phytoremediation of Chromium through Sunflower Plants Irrigated with Tannery Wastewater

**DOI:** 10.3390/plants9030380

**Published:** 2020-03-19

**Authors:** Ali Imran Mallhi, Shahzad Ali Shahid Chatha, Abdullah Ijaz Hussain, Muhammad Rizwan, Syed Asad Hussain Bukhar, Afzal Hussain, Zahid Imran Mallhi, Shafaqat Ali, Abeer Hashem, Elsayed Fathi Abd_Allah, Mohammed Nasser Alyemeni, Parvaiz Ahmad

**Affiliations:** 1Department of Applied Chemistry, Government College University Faisalabad, Faisalabad 38000, Pakistan; imranmallhi72@gmail.com; 2Department of Chemistry, Government College University Faisalabad, Faisalabad 38000, Pakistan; Chatha222@gmail.com (S.A.S.C.); abdullahijaz@gcuf.edu.pk (A.I.H.); 3Department of Environmental Science and Engineering, Government College University Faisalabad, Faisalabad 38000, Pakistan; mrazi1532@yahoo.com (M.R.); afzaalh345@gmail.com (A.H.); zahid.mallhi@yahoo.com (Z.I.M.); 4Department of Agronomy, Bahauddin Zakariya University, Multan 60800, Pakistan; asadbukhari@bzu.edu.pk; 5Department of Environmental Sciences, The University of Lahore, Lahore 54000, Pakistan; 6Department of Biological Sciences and Technology, China Medical University, Taichung 40402, Taiwan; 7Botany and Microbiology Department, College of Science, King Saud University, P.O. Box. 2460, Riyadh 11451, Saudi Arabia; habeer@ksu.edu.sa (A.H.); mnalyemeni@gmail.com (M.N.A.); 8Mycology and Plant Disease Survey Department, Plant Pathology Research Institute, ARC, Giza 12511, Egypt; 9Plant Production Department, College of Food and Agricultural Sciences, King Saud University, P.O. Box. 2460, Riyadh 11451, Saudi Arabia; eabdallah@ksu.edu.sa; 10Department of Botany, S.P. College, Srinagar, Jammu and Kashmir 190001, India

**Keywords:** chromium, wastewater, sunflower, biomass, chlorophyll contents

## Abstract

Heavy metals are rapidly polluting the environment as a result of growing industrialization and urbanization. The presence of high concentrations of chromium (Cr), along with other pollutants, is widespread in tannery wastewater. In Pakistan, as a result of a severe shortage of irrigation water, farmers use tannery wastewater to grow various crops with a consequent decline in plants’ yield. This experiment was performed to assess growth revival in sunflower plants irrigated with 0%, 25%, 50%, 75%, and 100% tannery wastewater, by foliar application of 0, 2.5, and 5.0 mM citric acid (CA). The wastewater treatment curtailed biomass accumulation, the growth rate, and chlorophyll contents by exacerbating the oxidative stress in sunflowers. Foliar application of CA considerably alleviated the outcomes of Cr toxicity by curbing the Cr absorption and oxidative damage, leading to improvements in plant growth, biological yield, and chlorophyll contents. It is concluded that foliar application of CA can successfully mitigate the Cr toxicity in sunflower plants irrigated with tannery wastewater.

## 1. Introduction

Kasur, a city in Punjab Pakistan, is famous for tanneries. According to estimates, around 144,502 to 215,036 gallons of wastewater day^−1^ is discharged into the environment by 650 registered tanneries [1]. Tannery wastewater contains several organic and inorganic pollutants [2]. Farmers mostly use tannery wastewater in periurban areas due to a scarcity of irrigation water [3,4]. There are different heavy metals in tannery wastewater, such as Cr, Cd, Mn, Pb, Fe, Ni, and Cu. Release of industrial effluent from tanneries is injurious for living organisms, including plants and animals [5]. The concentration of Cr in such water is considerably higher than other heavy metals [2]. Among heavy metals, chromium is ranked the 14th most noxious heavy metal globally; among different Cr oxidation states, Cr^+6^ is the most toxic because of the higher mobility/solubility and Cr^+3^ is the least toxic [6]. Plant growth and biomass is negatively affected through the application of industrial effluents. Outcomes of previous studies show that plant growth is negatively affected by 5 mg L^-1^ Cr in the nutrient solution [7]. Application of chromium (Cr^+6^) contaminated wastewater significantly reduced plant growth and photosynthesis in different crops, like maize, wheat, and sunflower, by causing oxidative stress [8,9,10]. The physiochemical processes of plants are badly affected by chromium stress [11]. Plant growth is reduced as a result of disruption in the photosynthetic process, damage to the ultrastructure of plant cells, oxidative stress such as electrolyte leakage (EL), hydrogen peroxide (H_2_O_2_), malondialdehyde (MDA) and changes in miRNAs and proteins [8,12,13,14,15]. Sunflower is an oil seed crop in Pakistan and a key source of edible oil worldwide. As a hyperaccumulator plant, it can be grown under stressful conditions such as under heavy metal stress. Sunflower plants were cultivated on 82,000 hectares of land, producing 40,000 tons of oil during 2017–2018 (Pakistan Economic Survey 2017–2018). Though the sunflower plant is considered metal resistant, toxic concentrations of Cr may negatively influence its growth and development. The outcomes of the previous investigations have revealed that Cr stress induced oxidative damage to plants with a consequent decline in growth and yield [16]. As a result of limited resources and exorbitantly increasing population pressure, contamination of land and water bodies with heavy metals, especially Cr, is a potential threat to food security and safety. The current situation demands an urgent remedy to ensure the provision of quality food to the population of Pakistan.

The beneficial effects of citric acid (CA) upon the uptake of Cr as well as in growth regulation in several plant species are well documented. The CA foliar application improved the germination rate and root weight of sunflower plant by improving the activities of several antioxidants enzymes including superoxide dismutase (SOD), catalase (CAT), peroxidase (POD) and ascorbate peroxidase (APX) [17]. Foliar application of CA, combined with 5-aminolevulinic acid (ALA), considerably increased the yield of sunflower plants. Citric acid spray on radish leaves reduced the uptake of Cr and its mobility [17]. Citric acid also increased the antioxidant enzyme activities, chlorophyll fluorescence, and reduced lipid peroxidation in *Brassica napus* [18]. Citric acid alleviated the Cr (Cr^+6^) toxicity by stimulating the antioxidant defense system in sunflower plants [19].

In the current study, it was hypothesized that CA may alleviate Cr toxicity in sunflower crop irrigated with tannery wastewater. Thus, the current study was designed to explore CA effects on morophogical, physiological, and chemical attributes of sunflower plants irrigated with tannery wastewater containing total Cr concentrations of 329 mg L^−1^.

## 2. Results

### 2.1. Plants Growth and Biomass

Growth features of sunflower plants under various treatments of tannery wastewater, along with application of CA, are presented in Figure 1 and Figure 2. Findings revealed that increasing concentration of wastewater treatment progressively reduced the plant height, root length, number of leaves per plant, and the leaf area (Figure 1), along with the fresh and dry weight of root, shoot, and leaves of sunflower (Figure 2). The highest decrease in the abovementioned parameters was noticed upon 100% wastewater treatments. However, CA application remarkably improved all the studied agronomic traits, both under stressed and normal growth environments. The maximum ameliorative effect of CA was observed at 5.0 mM foliar application treatment.

### 2.2. Photosynthetic Pigments

The application of wastewater severely decreased Chlorophyll *a, b,* and the total chlorophyll (Figure 3). Citric acid treatment significantly improved the pigment contents under Cr stress, as well as normal growth conditions. However, 5.0 mM proved to be the most effective concentration of CA in mitigating the detrimental effects of wastewater on photosynthetic pigments.

### 2.3. Oxidative Stress Parameters

The presence of Cr in tannery effluent resulted in oxidative damage to sunflower plant (Figure 4). By increasing wastewater concentration, it sharply elevated the MDA concentration in the leaves and roots of sunflowers. The maximum value for MDA contents was recorded under 50% and 100% wastewater treatment in leaves and roots, respectively. Foliar treatment of CA significantly decreased the lipid peroxidation as revealed by reduced MDA content in plants. The most conspicuous decline in MDA content was detected in plants as compared to respective control plants under a 75% and 100% wastewater treatment along with a 5.0 mM CA application in leaves and roots, respectively.

Similarly, wastewater treatment also caused oxidative damage by enhancing production of H_2_O_2_ in roots and leaves of plants, with the most pronounced effect at 50% wastewater application. Citric acid noticeably alleviated the oxidative damage to plants by decreasing H_2_O_2_ content both in the roots and leaves. The highest ameliorative effect was found in the plants irrigated with a 50% wastewater treatment in combination with a foliar application of 5.0 mM CA. Electrolyte leakage also exhibited a proportional rise with an increasing concentration of wastewater treatment, both in the leaves and roots of plants. Nonetheless, 5.0 mM CA considerably mitigated the consequences of stress as depicted by the reduction in EL in both parts of the plants.

### 2.4. Antioxidant Enzymes

Activities of SOD were increased with the 25% wastewater treatment and decreased thereafter with a further increase in stress level, both in the leaves and root of plants. The CA treatment enhanced SOD activities in sunflower leaves irrigated with normal tap water. Application of 2.5 mM CA enhanced the SOD activities, while 5.0 mM CA declined the SOD activities in sunflower plants irrigated with 25%–50% wastewater (Figure 5). Foliar applied CA notably improved leaf SOD activity under a 75%–100% wastewater application. In comparison with control plants, the SOD activities in roots increased considerably with a 25%–50% wastewater treatment, showing a consequent decline with a further increase in stress level. Citric acid application enhanced SOD activities in sunflower roots, with the highest improvement being observed in the control and 100% wastewater application supplemented with 2.5 mM and 5.0 mM CA treatment, respectively.

Mild to moderate levels (25%–50%) of wastewater treatment significantly decreased, while higher concentrations (75%–100%) increased the activities of POD in sunflower leaves. Application of CA significantly increased POD activities in sunflower leaves with a substantial surge at 100% wastewater irrigation, as compared with stress treatment alone. In roots, the activities POD increased with a 25% wastewater treatment and decreased with the increasing wastewater concentration. Citric acid application prominently improved the activities of POD in roots of sunflowers with the applied stress of tannery wastewater, in comparison with the stressed plants without CA treatment.

Wastewater treatment increased CAT activities at mild stress levels (25%) with a subsequent decreasing trend, both in the leaves and roots. However, CA supplementation further enhanced CAT activities in normal, as well as stressed plants in both plant parts. The effect of 5.0 mM CA concentration was the most prominent in improving CAT activities. Activities of APX were increased with a 25%–50% wastewater treatment, with a subsequent decrease under higher stress levels in both plant parts, i.e., leaves and roots. However, a 5.0 mM CA application significantly improved the APX activities of plants at all levels of wastewater treatment, as compared with those receiving no CA treatments.

### 2.5. Chromium Concentration

Concentration of Cr proportionally increased in sunflower root, stem, and leaves with the increasing wastewater concentration. Comparing with the control, the concentration of Cr in sunflower leaves increased by 97, 171, and 234 times, under 25%, 50%, 75%, and 100% wastewater treatments, respectively. Application of 5.0 mM CA significantly increased Cr concentration in sunflower leaves under all stress levels, with a conspicuous rise (35%) with the 100% wastewater treatment (Figure 6). The chromium concentration in the sunflower stem was increased by 92, 132, 179, and 208 times, under 25%, 50%, 75%, and 100% wastewater treatments, respectively. Exogenous CA application further increased the buildup of Cr in the sunflower stem under various concentrations of tannery wastewater. The maximum increase (36%) in stem Cr concentration was detected under a 100% tannery wastewater treatment along with a 5.0 mM CA application.

Heavy metals primarily affected roots of the plants under stressful conditions. In this study, the roots of stressed plants revealed comparatively higher Cr concentrations than stems and leaves. In comparison with the control, root Cr concentration was increased by 113, 211, 308, and 377 times, under 25%, 50%, 75%, and 100% wastewater treatments, respectively. Exogenously applied CA further increased the root Cr contents at all levels of wastewater application. The maximum value for Cr concentration in sunflower root was recorded with application of 5.0 mM CA with a 100% wastewater treatment.

## 3. Discussion

The present study elucidated the role of CA in mitigating the Cr toxicity caused by tannery wastewater in sunflower seedlings. Besides traces of several other heavy metals, tannery wastewater holds very high concentrations of Cr [2,20]. Various morphological parameters of sunflowers, such as biomass, leaf area, number of leaves per plant, root length, and plant height considerably decreased with an increasing wastewater concentration (Figure 1 and Figure 2), which may be attributed to the noxious amount of Cr in wastewater [20]. Previous reports have also revealed significant reductions in morphological features of various plant species under Cr toxicity [11,20]. Chromium induces morphological changes due to competition with other necessary nutrient elements, distortion of root and leaf ultrastructure, disruption in photosynthesis, and oxidative damage [12,14,21]. The findings of the current study are supported by the previous reports that Cr stress significantly decreased plant biomass, root length, and plant height in different plant species such as wheat, maize, tobacco, and spinach [11,14,19,20]. Citric acid treatment considerably improved the biomass and growth of stressed plants, in contrast to those irrigated with various wastewater concentrations without exogenous CA application (Figure 1 and Figure 2). Improvement in the morphological characteristics of the plants might be attributed to the enhanced absorption of vital nutrients by sunflower plants [22].

It is a well-established fact that photosynthetic pigments perform a very crucial role in a plants’ life due to their light harvesting function. Maqbool et al. [20] reported that Cr stress significantly affected the physiochemical attributes of the plants. In the current study, wastewater treatment prominently decreased the photosynthetic pigments in sunflower plants (Figure 3). Chlorophylls a, b, and total chlorophyll abruptly reduced with increasing tannery wastewater concentration. Reduction in plant growth may be attributed to the occurrence of high Cr concentrations in wastewater, which may have caused the structural impairment to chloroplast [11] and surplus generation of ROS under Cr induced stress [23]. Moreover, up-regulation of chlorophyllase under heavy metal stress constantly leads to an enormous reduction in chlorophyll pigments [24]. Citric acid application increased the chlorophyll contents in plants under tannery wastewater treatment. Similarly CA improved chlorophyll contents of wheat leaves under severe Cr toxicity [19]. Further, CA application also enhanced chlorophyll concentration in maize plants under drought condition [25]. Foliar application of CA caused an immense reduction in ROS generation and damage to the chloroplast, which resulted in an improvement in the pigment contents in plants [26].

Increased MDA content is an indication of membrane damage [27]. In the current study, H_2_O_2,_ MDA contents, and EL, were considerably enhanced in plant root and leaves of sunflower plants irrigated with tannery wastewater (Figure 3). However, CA treatment markedly ameliorated the lipid peroxidation and membrane damage by scavenging the free radicals and reducing ROS production [19,25].

Antioxidant enzymes perform a very significant role in protecting the plants from oxidative stress. The present study showed enhancement in SOD, POD, CAT, and APX activities at slight to moderate stress levels. Increasing doses of wastewater drastically reduced POD and SOD activities, except POD activity in the leaves (Figure 4). Similar findings were also reported previously [28]. Mild levels of Cr stress increased antioxidant enzyme activities, whereas serious stress reduced maize antioxidant enzyme activities due to relentless oxidative injury [11]. This could be because Cr toxicity alarms the antioxidant machinery of plants into starting to scavenge the ROS. Nevertheless, higher Cr levels suppress the antioxidant system due to a continuous and increased production of ROS.

Increasing concentrations of tannery wastewater drastically increased Cr concentration in sunflower root, stems, and the leaves of sunflower plants (Figure 6). Roots exhibited a much higher amount of Cr as compared to stem and leaves of the sunflower. Similar results were observed in rice and oil-seed rapes [11,29]. This might be due to Cr immobilization by sugars as macromolecules [19], followed by compartmentalization in root cell vacuoles [30]. Exogenous application of CA promoted uptake and accumulation of Cr towards different plant parts under various levels of tannery wastewater (Figure 6). However, Ali et al. [19] observed that CA application significantly decreased the uptake of different metals and their translocations to upper parts of various plants. The CA-mediated restricted absorption of Cr, due to the protection of the membrane system of plants, and resultant increase in the uptake of essential nutrients, might be the possible reason for enhanced plant growth [31]. Organic substances like fulvic acid and humic acid are well known for their constructive effects on heavy metals’ mobility and bioavailability, because they formulate composite organo-metal complexes [32,33,34]. Citric acid might have made complexes with Cr ions, which could be among many different possible reasons for the increase in Cr uptake by sunflower [22].

Crops may also have the ability to reduce Cr^6+^ to Cr^3+^, which would likely to happen in roots with the help of Cr reductase enzymes, same as those present in bacteria for detoxification of heavy metals; however, such enzymes have not been identified in plants [35]. The effects of Cr on plants vary with growth medium as well as different Cr oxidation forms [36]. It is well reported that Cr (VI) has more toxic effects towards plants compared to Cr (III) [37]. Riaz et al. [36] reported that wheat growth was mainly regulated by a different oxidation form of Cr than that of the total Cr. 

## 4. Materials and Methods

### 4.1. Experimental and Growth Conditions

Experiments were run at Botanical Garden of the Government College University, Faisalabad. The seeds of sunflower plants (variety Hysun-33) were carefully surface sterilized with 3% H_2_O_2_ for 20 min, followed by washing cautiously with distilled water. Soil was air dried and sieved through a 2 mm strainer. After that, five seeds were sown in each pot filled with 8 kg of soil. After two weeks of germination, thinning was done and only two plants were kept in each pot. Prestudy soil analyses were performed according to the established protocols, as described in Table 1. Soil texture, sodium adsorption ratio (SAR), electrical conductivity (EC), soluble ions, and different trace elements were determined by Bouyoucos [38], Page et al. [39], US Salinity Lab. Staff [40], and Soltanpour [41], respectively.

### 4.2. Analysis of Tannery Wastewater and Application of Treatments

Tannery wastewater was analyzed for various physicochemical characteristics through standard protocol [42] (Table 2). Two weeks old sunflower plants were irrigated every third day with five different levels (0%, 25%, 50%, 75%, and 100%) of tannery wastewater until harvesting. The number of pots were considered as one replicate per percentage of wastewater. After 1 week of the wastewater irrigation, the citric acid was sprayed at different concentrations (0, 2.5 and 5.0 mM), with a 2-day interval, throughout the experiment. The plants in the control group were cautiously sprayed with the same quantity of distilled water. Chemical fertilizers were applied as described previously [19].

### 4.3. Measurement of Morphological Attributes

After 8 weeks of the 1st CA treatment, the plants were harvested. After harvesting, the plants were washed with distilled water and were carefully separated into stem, leaves, and roots. Data on various agronomic traits, such as plant height, root length, fresh and dry weight of roots and shoots, number of leaves per plant, and leaf area, were measured according to the standard procedures.

### 4.4. Assessment of Oxidative Stress

Oxidative stress was assessed by determining EL, MDA content, and H_2_O_2_ concentration, according to previously described methods [43,44,45].

### 4.5. Photosynthetic Pigments and Antioxidant Enzyme Analysis

The contents of chlorophyll *a*, *b,* and total chlorophyll were assessed by following established protocol [46]. Superoxide dismutase and POD activities were examined according to Zhang et al. [47]. Activities of CAT and APX were quantified by using the already developed protocol [48].

### 4.6. Determination of Cr Contents

Plant samples were analyzed for Cr content in leaves, stems, and roots following the protocol described by Ehsan et al. [49]. Samples (0.5 g) were ground into fine powder and burnt to ashes in a muffle furnace at 1000 °C for 12 h, followed by acid digestion overnight. Afterwards, digested samples were filtered several times to get a clear extract. Finally, samples were analyzed by atomic absorption spectrophotometer (Halo DB-20/DB-20S, Dynamica Company, London, UK) and total Cr concentration was calculated by drawing standard curve.

### 4.7. Statistical Analysis

Complete randomized design was applied along with 3 replicates. Analysis of variance (ANOVA) was applied using statistical software (SPSS, version 23.0 for windows; IBM Corporation, Armonk, New York, U.S). A post hoc test followed by a Duncan test was applied to see significant difference among different treatments.

## 5. Conclusions

This study demonstrated that wastewater application reduced the morphological and photosynthetic attributes of plants. Substantial amount of total Cr accumulated in sunflower when provided with tannery wastewater. The CA application enhanced growth and photosynthetic pigments of sunflowers by decreasing the oxidative damage. Moreover, CA increased activities of antioxidant enzymes in sunflower plants. The total Cr uptake was enhanced with enhancing the concentration of CA. Current findings suggest that citric acid application could be an easy and effective strategy to alleviate chromium toxicity. However, further investigations should be done in the future on a molecular level with detailed mechanistic approaches in this regard.

## Figures and Tables

**Figure 1 plants-09-00380-f001:**
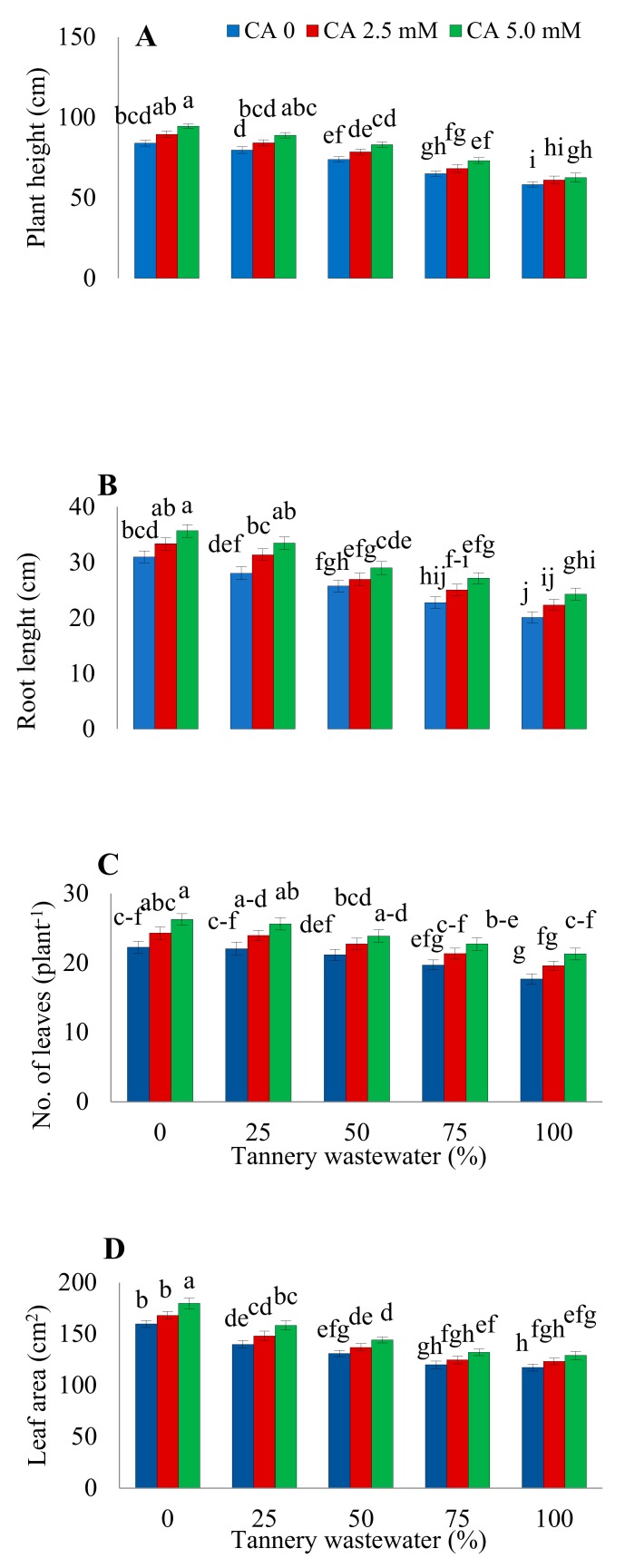
Impact of wastewater and citric acid (CA) on sunflower height (**A**), root lengths (**B**), number of leaves per plant (**C**), and leaf area (**D**) of sunflower plants. Data are means of 3 independent replicates, and different lettering indicates a significant difference among the values at *p* ˂ 0.05.

**Figure 2 plants-09-00380-f002:**
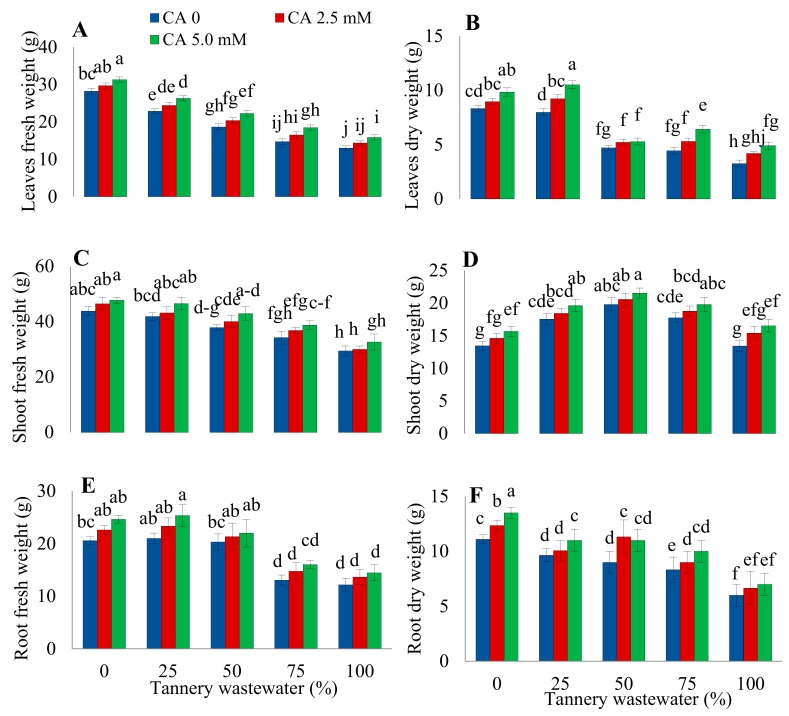
Impact of wastewater and citric acid (CA) treatments on leaf fresh weight (**A**), leaf dry weight (**B**), shoot fresh weight (**C**), shoot dry weight (**D**), root fresh weight (**E**) root dry weight (**F**) of sunflower plants. Data are means of 3 independent replicates, and different lettering indicates a significant difference among the values at *p* ˂ 0.05.

**Figure 3 plants-09-00380-f003:**
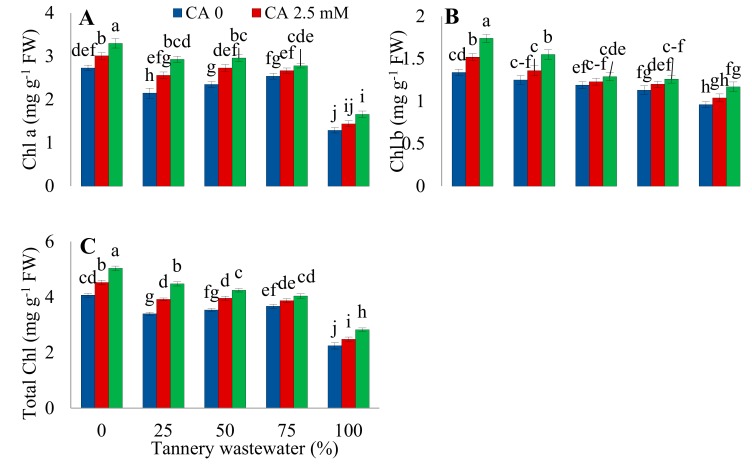
Impact of wastewater and citric acid (CA) treatment on chlorophyll a (**A**), chlorophyll b (**B**), and total chlorophyll (**C**) of sunflower plants. Data are means of 3 independent replicates and different lettering indicates a significant difference among values at *p* ˂ 0.05.

**Figure 4 plants-09-00380-f004:**
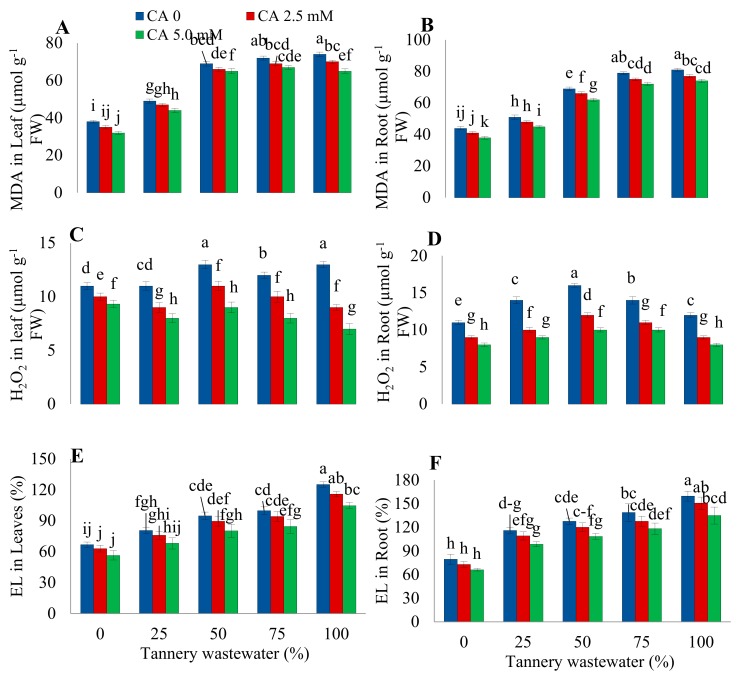
Impact of wastewater and citric acid (CA) treatment on leaf malondialdehyde (MDA) (**A**), roots MDA (**B**), leaves H_2_O_2_ (**C**), root H_2_O_2_ (**D**), leaf electrolyte leakage (EL) (**E**), and root EL (**F**) of sunflower plants. Data are means of 3 independent replicates, and different lettering indicates a significant difference among the values at *p* ˂ 0.05.

**Figure 5 plants-09-00380-f005:**
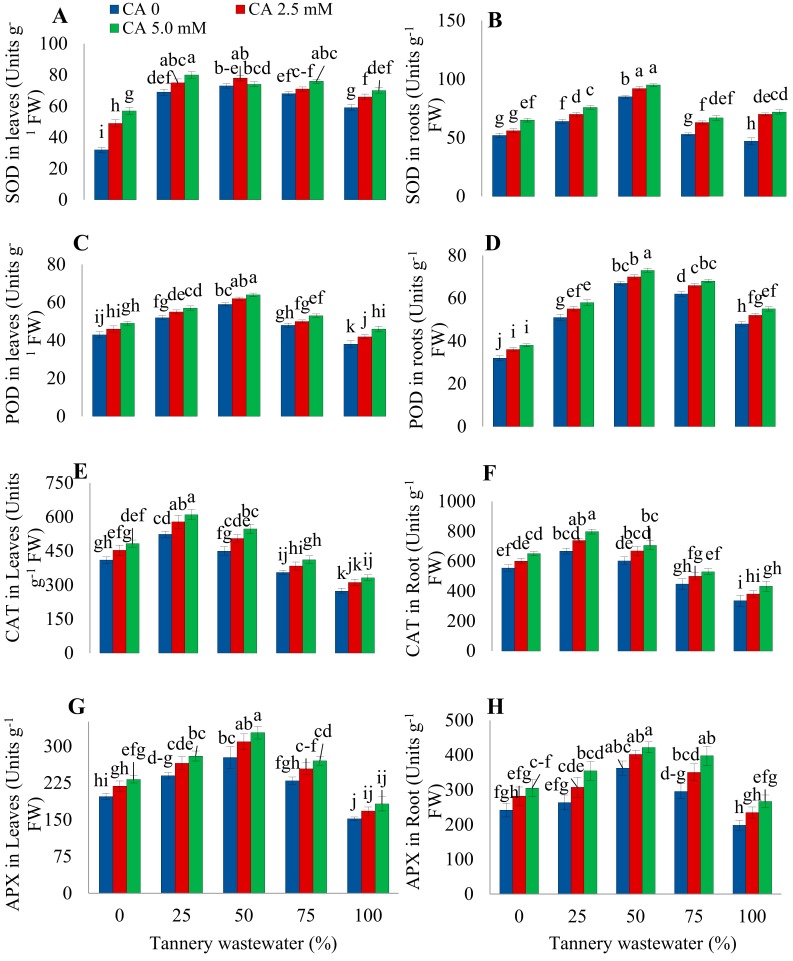
Impact of wastewater on and citric acid on leaf SOD (**A**), root SOD (**B**), leaf POD (**C**), root POD (**D**), leaf CAT (**E**), root CAT (**F**), leaf APX (**G**), and root APX (**H**) of sunflower plants. Data are means of 3 independent replicates, and different lettering indicates a significant difference among values at *p* ˂ 0.05.

**Figure 6 plants-09-00380-f006:**
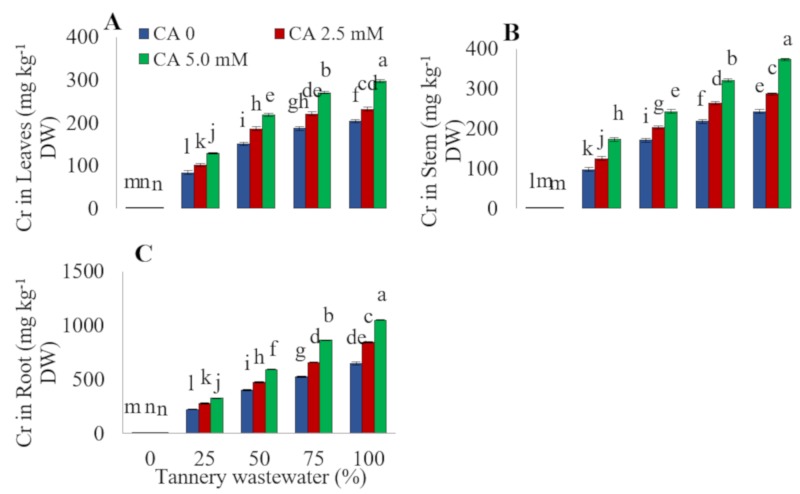
Impact of wastewater and citric acid on Cr concentration in leaves (**A**), stems (**B**), and roots (**C**) of sunflower plants. Data are means of 3 independent replicates and different lettering indicates a significant difference among the values at *p* ˂ 0.05.

**Table 1 plants-09-00380-t001:** The physicochemical characteristics of the soil used in the pot study.

Texture	Sandy Loam
Silt	15.0%
Sand	67.9%
Clay	17.10%
EC	1.96 dS m^−1^
pH	7.61
SAR	1.89 (mmol L^−1^)^1/2^
Available P	2.11 mg kg^−1^
Organic matter	0.59 %
HCO_3_^−1^	2.51 mmol L^−1^
SO_4_^−2^	11.44 mmol L^−1^
Cl^-^	5.45 mmol L^−1^
Ca^2+^ + Mg^2+^	13.98 mmol L^−1^
K^+^	0.04 mmol L^−1^
Na^+^	5.23 mmol L^−1^
Available Zn ^2+^	0.77 mg kg^−1^
Available Cu ^2+^	0.31 mg kg^−1^
Available Cr	0.16 mg kg^−1^

Here in Table 1, the EC stands for electrical conductivity and SAR stands for sodium adsorption ratio.

**Table 2 plants-09-00380-t002:** Characteristics of tannery wastewater used for irrigation.

Parameters	Values
COD	2897 mg L^−1^
BOD	876 mg L^−1^
TOC	969 mg L^−1^
Oil & grease	11 mg L^−1^
pH	4.13
EC	91.8 dS/m
TDS	64,968
Total Cr	329 mg L^−1^
K^+^	41 mg L^−1^
Carbonate	ND
Ca^2+^+Mg^2+^	3.1 mmol_c_ L^−1^

Here in Table 2, COD stands for chemical oxygen demand, BOD stands for biological oxygen demand, TOC stands for total organic compounds, EC stands for electrical conductivity, TDS stands for total dissolved solids, and ND stands for not detected.

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
