# Peer review of "Citric Acid Assisted Phytoremediation of Chromium through Sunflower Plants Irrigated with Tannery Wastewater"

_plants, 2020, doi:10.3390/plants9030380_

Round 1

Reviewer 1 Report

The first concern is about the originality of the present work. The authors in the introdcution mention soem papers fro the literature that have used CA for this purpose but do not specify what is the new in their work, apart from using thsi regime to test if it works. Another issue in the work, is that the authors always refer to chromium, while all literature when it deals with chromium discriminates between Cr6 and Cr3. It is very strange that the authros do not even mention the speciation of chromium in wastewaters. The figures are very difficult to be read and understood. Very small, too many together. I also found strange, that the materials and methods sections comes last, while it woule be useful to have it before the results, in order to read how the measurements have been done and what eevidence do we have about each factor. Overall the paper is of low quality, with low innovation and I do not suggest publication in its present form.

Author Response

The first concern is about the originality of the present work. The authors in the introduction mention some papers from the literature that have used CA for this purpose but do not specify what is the new in their work, apart from using this regime to test if it works. Another issue in the work, is that the authors always refer to chromium, while all literature when it deals with chromium discriminates between Cr6 and Cr3. It is very strange that the authors do not even mention the speciation of chromium in wastewaters. The figures are very difficult to be read and understood. Very small, too many together. I also found strange, that the materials and methods sections comes last, while it would be useful to have it before the results, in order to read how the measurements have been done and what evidence do we have about each factor. Overall the paper is of low quality, with low innovation and I do not suggest publication in its present form

Response

Thank you very much for giving us a chance to respond to the comments. We have add hypothesis and have revised our objective clearly in order to show the novelty of our work. Please see lines 77-79.

The reviewer talked about Cr speciation. We do not have done our work on speciation in this study but we will consider this nice suggestion in our upcoming future study, so thanks for nice suggestion. We have improved the figures in our revised manuscript.

According to journal format/style the materials and methods section comes in last than that of the results section, so we have just followed the journal instructions.

Reviewer 2 Report

Please find my comments in the attached file.

Author Response

Reviewer 2

General Comments

Although the use of citric acid was already reported to improve plant development and heavy metal uptake in other plant species, the authors present here the use of CA to improve sunflower growth watered with wastewater of tanneries contaminated with Cr.

The data presented show that CA also increase Cr uptake and accumulation in maize. Therefore, CA could be used for decontaminating the soils watered with tannery wastewater. However, I think that it should be stressed, in the discussion or at least in conclusion, that CA absolutely should not be used for growing sunflower plants that are meant to be a source of food.

Response

Thank you very much for giving us a chance to respond to the comments. We have made every effort to address reviewer’s concerns as clearly and succinctly as possible, as demonstrated in the following pages.

Comments for the Introduction

The introduction is incomplete and needs more explanations about the plant physiology/oxidative stress and should include the definitions of MDA, EL, POD, CAT, APX.

Response

Thanks for nice suggestion. We have made changes and highlighted as suggested and have defined the MDA, EL, POD, and APX as well in our revised manuscript.

Comments for the materials and methods

Methods: All the abbreviations should be explained and statistics analysis should be described. The number of pots considered as one replicate per percentage of wastewater should be mentioned.

Response

We have described the statistical analysis as per nice suggestion, please see lines 297-300. Also, we have mentioned the number of pots considered as one replicate per percentage of wastewater in revised manuscript.

Line 241: “attentively” should be removed

Response

We have removed the word “attentively” as suggested.

Line: 242: “thining” do the authors mean “trimming”?

Response

No it’s thinning, which means to reduce the numbers of plants per pots by uprooting the extra plants and crushing them into same pot’s soil. 

Title table 1: “phisico” instead of “physic”

Response

Thanks for catching this. We have corrected it in revised version.

Table 1: Please precise what EC and SAR are

Response

EC stands for electrical conductivity and SAR means Sodium adsorption ratio. We have mention these below Table 1 in our revised manuscript.

Line 249: If a special handling was performed, please precise it, otherwise “carefully” should be removed

Response

Thank you, we have removed the word ‘carefully’.

Table 2: Please explain all the abbreviations used in the Table 2 and what means “Nil”.

Response

We have explained all the abbreviations as suggested and ‘Nil’ has been replaced by ‘ND’ means not detected. Also we have mentions all these below Table 2.

Line 272: “spectrophotometer (make and model of AA)”: please provide them

Response

Thanks for catching this, we have mention the spectrophotometer model in revised version.

Line 279: It should be concluded then that CA should not be used for sunflower plants supposed to serve as food.

Response

Thanks for nice comment, we have revised our conclusion.

Comments about the figures

The organization of the graphs has to be improved: for each figure, as the X axis is the same for all the graphs, all the graphs should have same size and be aligned within each figure

Response

We have made changes as per nice suggestion.

Figure 1: “chart area” label should be removed

Response

Thanks for catching this, we have removed in revised version.

Figure 1D: Can the authors explain how they can get more than 6 plants per pot, whereas according to the methods only 5 seeds were added per pot? Figure 1D: Should be commented in the text, otherwise removed

Response

Thanks for nice comment. We have removed Fig 1D in the revised manuscript.

Figure 2: Can the authors explain why the shoot fresh weight and the shoot dry weight don’t present the same tendencies with increasing wastewater?

Response

Thanks for nice comment. There is may be the difference of water contents in fresh and dry weight of shoot. This might be the reason for varying fresh and dry weight.

Figure 4: Different colors should be used for the 3 different concentrations of CA

Response

Thanks for nice suggestion, we have made changes in our revised manuscript as suggested.

More Comments:

Line 28: Please remove the extra dot

Response

Thanks for catching this, the extra dot has been removed.

Line 65: The definition of CA should be added.

Response

CA definition has been added in the revised manuscript.

Line 66: A reference should be provided for the CA effect on sunflower

Response

A reference has been provided as suggested.

Line 66: remove the isolated “s”

Response

Thanks, we have removed that.

Line 67: definition of ALA should be added in the text

Response

We have added the ALA definition as suggested.

Line 71: Can the author’s double check the reference please?

Response

We have double checked the references considering your nice suggestion.

Line 110: “most conspicuous decline in MDA contents” please precise relative to what

Response

Thanks for catching, we have completed the sentence by relating stressed plants MDA contents comparing with respective control plants.

Line 141: the sentence could be more explicit

Response

Thank you very much, the sentence has been made more explicit as suggested.

Line 205: This statement is not in agreement with the results shown in Fig 6. On the contrary, CA increased the concentration of Cr in aerial parts.

Response

Thanks for nice comment. We have removed that line from the revised manuscript.

Line 209: This should be already explained in the introduction

Response

We have made the change and removed that explanation.

Line 226-227: It would be useful to precise here if other studies mentioned an increased uptake and accumulation of Cr after CA treatment

Response

Thanks for nice comment. Actually, before this sentence, we have cited those study which has increased the Cr uptake. But, in this sentence we have mentioned a study with contrary results along with the possible reason.

Reviewer 3 Report

This paper reports information and procedure on the use of Phytoremediation of Cr(VI) by citric acid, for sunflower irrigated with tanneries wastewater. The paper may be accepted after major revision.

The paper can be accepted for publication in the Plants after the following suggestions and comments have been taken into account:

Pag.3 –Figure 1. Do graphs “C” and “E” relate to the same variable? That is: is the area of the leaves relative to the total number of leaves present in a single plant? If so, the area of the leaves and the number of leaves have the same trend; Nowhere in the work is it clear what the times of contact with citric acid are; the time intervals after which measurements and determinations (measurements of leaf weight, plant height, root length, and so on for all kind of determination) are made; The authors believe that all the effects observed on plants are related only to Cr. They cite numerous papers on the effects of chromium on plants. However, since the irrigation waters, more or less diluted, contain not only chromium but probably other metal ions and also other organic and / or inorganic substances, I recommend testing with aqueous solutions containing different concentrations of chromium with which to irrigate the plants and verify their effects, in the presence and not, of citric acid distributed on the leaves. 11 – line 272: Be careful to re-read the scientific work, in brackets perhaps you wanted to indicate the brand and model of the instrument used.

Author Response

Reviewer 3

This paper reports information and procedure on the use of Phytoremediation of Cr(VI) by citric acid, for sunflower irrigated with tanneries wastewater. The paper may be accepted after major revision.

The paper can be accepted for publication in the Plants after the following suggestions and comments have been taken into account:

Response

Thank you very much for giving us a chance to respond to the comments. We have made every effort to address reviewer’s concerns as clearly and succinctly as possible, as demonstrated in the following pages.

Pag.3 –Figure 1. Do graphs “C” and “E” relate to the same variable? That is: is the area of the leaves relative to the total number of leaves present in a single plant? If so, the area of the leaves and the number of leaves have the same trend; Nowhere in the work is it clear what the times of contact with citric acid are; the time intervals after which measurements and determinations (measurements of leaf weight, plant height, root length, and so on for all kind of determination) are made; The authors believe that all the effects observed on plants are related only to Cr. They cite numerous papers on the effects of chromium on plants. However, since the irrigation waters, more or less diluted, contain not only chromium but probably other metal ions and also other organic and / or inorganic substances, I recommend testing with aqueous solutions containing different concentrations of chromium with which to irrigate the plants and verify their effects, in the presence and not, of citric acid distributed on the leaves. 11 – line 272: Be careful to re-read the scientific work, in brackets perhaps you wanted to indicate the brand and model of the instrument used

Response

Thanks for valuable comments. In figure 1, graph ‘C’ and ‘E’ don’t relate the same variable. In graph ‘C’, there is counting of the numbers of leaves on a single plant while in graph ‘E’, it’s the leaf area which is different parameter from previous one. We have mentioned the contact time of citric acid and also have mentioned the time interval after that harvesting was made, in our revised manuscript, please see line 260 and line 269. If we talk about heavy metals concentrations in tanneries wastewater, there is major contribution of Cr and very minute concentrations of other heavy metals which don’t affect plant significantly. That’s why we didn’t consider other heavy metals. However, it was nice comment and we will consider this valuable suggestion in our upcoming future studies. For line 272, thanks for catching this, we have mentioned the instrument model in our revised manuscript. Please see line 294. 

Reviewer 4 Report

The manuscript examines the effect of 0, 2.5 and 5.0 32 mM citric acid foliar application on the growth, photosynthetic pigments and activities of antioxidant enzymes in sunflower plants irrigated with 0, 25, 50, 75 and 100 % tanneries wastewater enriched with Cr. An interesting work with satisfactory selection of methodologies and useful data to alleviate Cr toxicity. There are some minor comments:

Please correct all the figures. In many cases, the different letters to show statistical significance are not visible, making the results not clear.

Lines 225 Similar results were also found recently for lettuce plants.  Add the relevant reference https://doi.org/10.1016/j.chemosphere.2018.07.046

Author Response

Reviewer 4

The manuscript examines the effect of 0, 2.5 and 5.0 32 mM citric acid foliar application on the growth, photosynthetic pigments and activities of antioxidant enzymes in sunflower plants irrigated with 0, 25, 50, 75 and 100 % tanneries wastewater enriched with Cr. An interesting work with satisfactory selection of methodologies and useful data to alleviate Cr toxicity. There are some minor comments:

Response

Thank you very much for giving us a chance to respond to the comments. We have made every effort to address reviewer’s concerns as clearly and succinctly as possible, as demonstrated in the following pages.

Please correct all the figures. In many cases, the different letters to show statistical significance are not visible, making the results not clear.

Response

Thanks for nice comment. We have made changes in our revised manuscript as suggested.

Lines 225 Similar results were also found recently for lettuce plants.  Add the relevant reference https://doi.org/10.1016/j.chemosphere.2018.07.046 

Response

Thanks for nice suggestion. We have cited the relevant study in revised version as suggested. Please see reference 34.

Round 2

Reviewer 1 Report

The authors did not resopnd to my comments and neither made an effort to get into the important aspects of my comments. Chromium speciation is very important and they should have at least bothered to try to explain the reasons however, they saidd they will do it in the next work.

Author Response

Reviewer 1

The authors did not respond to my comments and neither made an effort to get into the important aspects of my comments. Chromium speciation is very important and they should have at least bothered to try to explain the reasons however, they said they will do it in the next work.

Response

All the authors are very thankful to the reviewers for critically reviewing the manuscript. We have differentiated between Cr3 and Cr6 in the introduction and in the discussion where applicable. We have further added the relevant literature in the text about the Cr speciation. The most of the studied published have used Cr6 for the purpose of reduction of its toxicity in plants as this form of Cr is toxic to living things than other Cr forms. Even we have not did speciation in our current study but we have talked about and highlighted the Cr speciation in both introduction and discussion sections. Please see lines 49-50 and lines 256-261

Reviewer 3 Report

The authors, all things considered, answered and implemented their work. Work can be accepted in this form.

Author Response

Reviewer 2

The authors, all things considered, answered and implemented their work. Work can be accepted in this form.

Response

Thank you for your positive comments. We appreciate the reviewer for spending the time to evaluate and providing thoughtful comments to improve our manuscript.

This manuscript is a resubmission of an earlier submission. The following is a list of the peer review reports and author responses from that submission.